# Visual representations derived from multiplicative interactions

**Eric Elmoznino**
Department of Cognitive Science
Johns Hopkins University
Baltimore, MD 21218
eric.elmoznino@gmail.com

**Michael F. Bonner**
Department of Cognitive Science
Johns Hopkins University
Baltimore, MD 21218
mfbonner@jhu.edu

## Abstract

Biological sensory systems appear to rely on canonical nonlinear computations that can be readily adapted to a broad range of representational objectives. Here we test the hypothesis that one such computation—multiplicative interaction—is a pervasive nonlinearity that underlies the representational transformations in human vision. We computed local multiplicative interactions of features in several classes of convolutional models and used the resulting representations to predict object-evoked responses in voxelwise models of human fMRI data. We found that multiplicative interactions predicted widespread representations throughout the ventral stream and were competitive with state-of-the-art supervised deep nets. Surprisingly, the performance of multiplicative interactions did not require supervision and could be achieved even with random or hand-engineered convolutional filters. These findings suggest that multiplicative interaction may be a canonical computation for feature transformations in human vision.

## 1   Introduction

Nonlinear transformations are central to the representational power of visual cortex. Through nonlinear transformations, visual cortex converts low-level sensory inputs into complex representations that directly support behavior. Researchers have long sought to understand the nature of these nonlinear transformations (DiCarlo, Zoccolan, & Rust, 2012). Deep learning in artificial neural networks is one promising approach because any complex nonlinear transformation can be approximated through multiple stages of elementary nonlinear operations (e.g., rectification) (Hornik, 1991). However, there is considerable evidence that even the most elementary nonlinear operations of cortical neurons may be far more complex than those of artificial neurons in conventional neural networks (Silver, 2010). Understanding these cortical nonlinear operations may reveal critical inductive biases and computational efficiencies of visual cortex that are not accounted for by current computational theories.

Here we explore the possibility that multiplicative interaction is a canonical nonlinear computation used in biological vision. In its simplest form, multiplicative interaction is a computation in which a neuron outputs the product of two or more input neurons. Previous work in machine learning has shown that multiplicative interaction can improve the expressivity, compactness, and learnability of multilayer perceptrons (Jayakumar et al., 2020). In neuroscience, it is known that multiplicative interaction has a specialized function in gain modulation, whereby a neuron's tuning profile is multiplicatively modulated by contextual factors, such as attention (Ferguson & Cardin, 2020). What is not known is whether multiplicative interaction serves a broader role in cortical computation by functioning as a canonical nonlinear transformation in sensory coding.

2nd Workshop on Shared Visual Representations in Human and Machine Intelligence (SVRHM), NeurIPS 2020.

We implemented a multiplicative-interaction layer for convolutional neural networks (CNNs) that computes second-order interactions between pairs of first-order feature maps output by convolutional filters. We used this computational architecture in combination with voxelwise encoding models of fMRI data to examine the power of multiplicative interactions for predicting object-evoked responses in human visual cortex. We found that the simple addition of a multiplicative-interaction layer to a CNN produced striking and widespread improvements in encoding-model performance across much of visual cortex. This was true for multiple classes of CNNs, including hand-engineered models that required no training as well as supervised models that were pre-trained for image classification. In fact, our models with multiplicative interaction layers were competitive with supervised conventional CNNs while using an order of magnitude fewer parameters. These findings suggest that multiplicative interaction may be a powerful and general-purpose nonlinear transformation that is capable of supporting diverse representational objectives in visual cortex.

## 2 Results

The general approach that we take in all of our experiments is shown in Figure 1 and can be summarized as follows:

1. Define a CNN architecture that takes an image as input and outputs a number of feature maps. Call this a first-order model.
2. Define a second-order model that computes the multiplicative interactions between each pair of feature maps from the first-order model.
3. Train a linear encoding model to predict fMRI activity from the output of the first-order model applied to the same images that subjects saw under the scanner. Do the same using the second-order model.
4. Compare the predictive accuracies of the two encoding models across a number of brain regions.

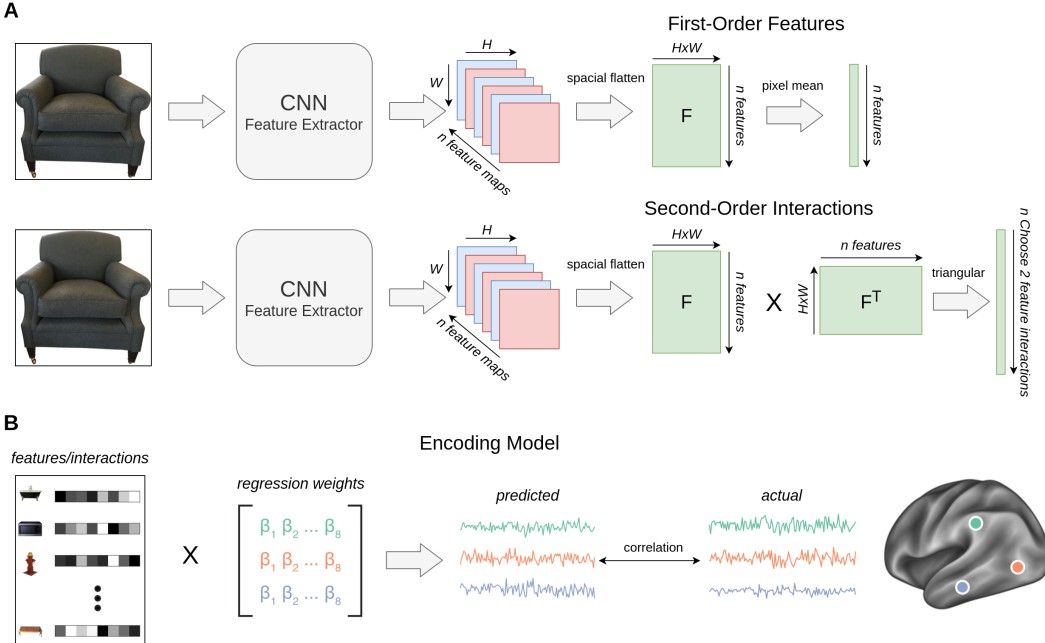

**Figure 1: Overview of our modeling approach. A)** We fed images through CNN feature extractors to output a set of first-order feature maps. To compute second-order interactions, we multiplied the flattened feature maps by their transpose and kept the upper triangular portion of the resulting matrix. **B)** We trained linear encoding models to predict fMRI responses in visual cortex from either first-order features or their second-order interactions and compared their accuracies.

If multiplicative feature interactions play a central role in neural processing as a canonical computation, then we should expect them to improve fMRI encoding accuracy over a wide range of feature sets. For this reason, we compared first- and second-order encoding models over features extracted from 5 different CNNs. Three of these consisted of 1-layer linear convolutions with hand-engineered filter banks. Two of the hand-engineered models are based on known representations of visual cortex: i) simple edge detectors tuned to orientation and ii) curvature detectors tuned to both orientation and degree of curvature. The third hand-engineered model consisted of random-weight filters, which can be surprisingly effective in extracting useful features (Cao, Wang, Ming, & Gao, 2018). Our two other feature extractors were deep CNNs trained on image classification tasks, and we used the feature maps output by the final convolutional layer to train the encoding models. All of these models are described in further detail in Appendix A.

There are several ways one could compute second-order multiplicative interactions from the feature maps output by our CNNs. Here, we took a simple approach by computing the products between every pair of feature maps then aggregating each of the resulting interaction maps across spatial dimensions by taking their sum. These two operations correspond to taking the matrix multiplication of the spatially flattened feature maps and their transpose and then keeping only the upper triangular section. Mathematically, this operation is given by the Equation 1:

$$M = triangular(FF^T) \tag{1}$$

Where $F$ is a $(n, hw)$ matrix of $n$ first-order feature maps flattened across the spatial dimensions, $triangular(\cdot)$ corresponds to flattening the upper triangular part of a matrix into a vector, and $M$ is a vector containing all $\binom{n}{2}$ pair-wise multiplicative interactions. This operation is described further in Appendix B. Since the above operation aggregates second-order feature interactions across spatial dimensions through summation, we performed a similar aggregation when fitting first-order encoding models by taking the spatial mean of the feature maps, resulting in a vector of size $n$.

With our feature extractors and second-order operation in hand, we wished to evaluate the utility of multiplicative feature interactions in models of human visual cortex. We approached this problem by fitting linear-regression encoding models that predicted fMRI responses to object images from the second-order interactions of features output by our CNNs. We then compared the predictive accuracy of these encoding models to their first-order counterparts, which instead predicted fMRI responses directly from the CNN features. Further details regarding our fMRI dataset and encoder training methodology can be found in Appendix 5 and in a preprint for another project using these data (Bonner & Epstein, 2020).

We found that second-order multiplicative interactions provided a significant improvement in fMRI voxelwise prediction accuracy, as shown in Figure 2. In support of our hypothesis that multiplicative interactions are a canonical computation widespread in human vision, the improvements resulting from multiplicative interactions were observed across all brain regions we tested. These included both low-level regions in early visual cortex as well as more high-level category-selective regions. Multiplicative interactions also improved prediction accuracy across all 5 feature extraction models considered, including both simple hand-engineered models and supervised deep CNN architectures trained on classification tasks. This suggests that whatever role these specific features might play in the representations of visual cortex, multiplicative interactions between these features might play an even more important one. In other words, the consistency with which multiplicative interactions improved the performance of diverse encoding models suggests that visual representations cannot be understood in terms of first-order feature sets alone.

Of particular note is the dramatic improvement observed in our ImageNet-trained CNN (bottom center). Using the first-order features from this CNN, neural activity was actually predicted quite poorly, achieving a mean Pearson correlation of less than 0.1 on the test set for all brain regions. This was expected, since the first-order model was intentionally designed to be a highly reduced version of AlexNet with a similar architecture but an order of magnitude fewer parameters. However, when using the exact same CNN and simply computing the second-order interactions of the extracted features, encoder performance improved to near noise ceiling levels for all brain regions considered (Appendix E). In addition, this model also matched the encoder performance of a conventional pre-trained AlexNet (Appendix F).

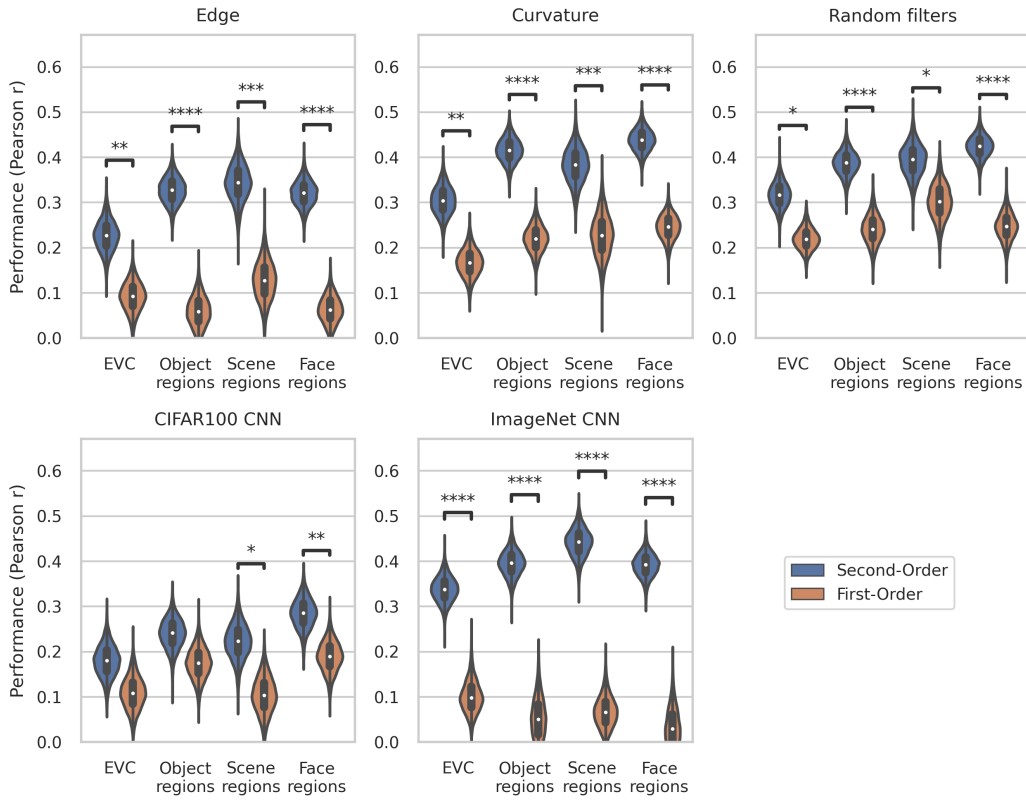

**Figure 2: Voxelwise encoding models of CNN features and their second-order interactions.** Second-order multiplicative interactions between CNN features were highly predictive of fMRI responses to objects in visual cortex. Their predictive value was significantly greater than that of the original first-order features used to compute them. This improvement was observed for all feature extraction models across all brain regions considered, including early visual cortex (EVC) and higher-level regions associated with object, scene, and face processing. Data are from 4 subjects viewing 810 images of real-world objects from 81 different categories. We fit predictive linear encoders of fMRI responses, and we measured performance as the mean voxelwise correlation between predicted and actual fMRI responses in a cross-validation design. Violin plots represent means and bootstrap distributions for each visual region. *\*\*p<0.01, \*\*\*p<0.001, \*\*\*\*p<0.0001*

It is also worth noting that in the above results, our supervised models were trained for classification *without* the second-order multiplicative interaction layer. The multiplicative interactions were only computed afterwards, when training the encoding models. If multiplicative interactions are computed in visual cortex, then a fairer comparison would be to include these computations when training the supervised models on their classification tasks. When we do this, fMRI encoding accuracy improves even further, as shown in Figure 3. In addition, we observed a substantial improvement in accuracy on the classification tasks when using the multiplicative interactions prior to fitting our fMRI encoders. For instance, our ImageNet-trained model achieved a top-1 classification accuracy of 14% on the test set over 1000 classes when using first-order features, but an accuracy of 27% when using second-order interactions. This improvement is interesting in and of itself, and more work can be conducted to see if these kinds of multiplicative interactions have practical use in deep learning architectures for computer vision.

**Controlling for increased dimensionality.** An alternative explanation of our results might be that computing the second-order multiplicative interactions of a set of features results in a substantial increase in the dimensionality of the input used to train our encoding models and, therefore, an increase in the number of regression parameters. We addressed this possibility with an experiment that used only a random subset of the second-order interactions, matching the number of first-order features that were used to compute the interactions. We performed this analysis for a wide range of first-order feature sizes, all using the random filter bank feature extractor so as to avoid having to

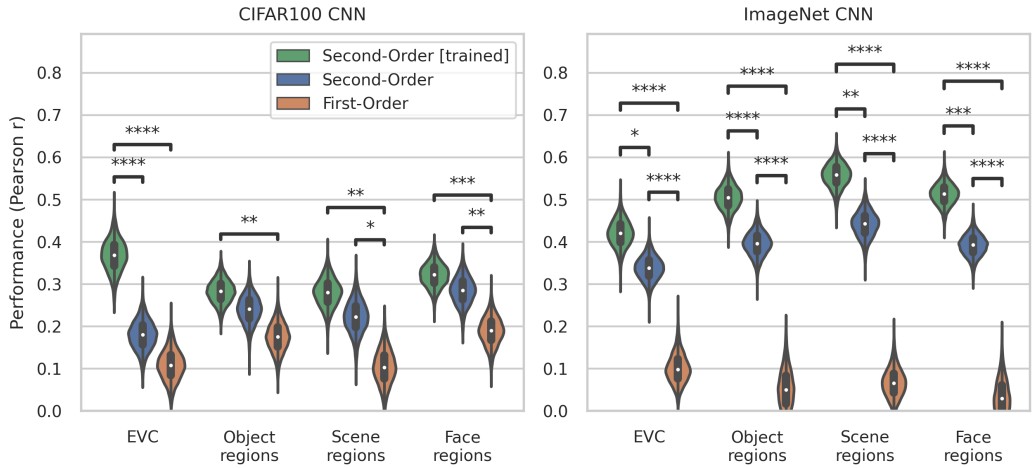

**Figure 3: Voxelwise encoding models of supervised CNN features and their second-order interactions applied during classification training.** Second-order interactions perform better as models of fMRI activity in visual cortex when they are used to train the CNN on its classification task (Second-Order [trained]). Violin plots represent means and bootstrap distributions for each visual region. *\*\*p<0.01, \*\*\*p<0.001, \*\*\*\*p<0.0001*

arbitrarily pick hyperparameters other than the number of filters used. The results for this analysis are shown in Figure 4. We found that beyond a dimensionality of roughly 10, encoders trained using a random subset of second-order multiplicative interactions performed significantly better than their first-order counterparts, despite the dimensionality being matched in the two cases. Furthermore, we saw that while the encoding accuracy of the first-order models saturated early as the number of features increased, the second-order models continued to improve. This suggests a possible explanation for our findings and a computational role for multiplicative interactions, namely that their representational power might scale better as a function of increased first-order feature dimensionality than that of the first-order features themselves.

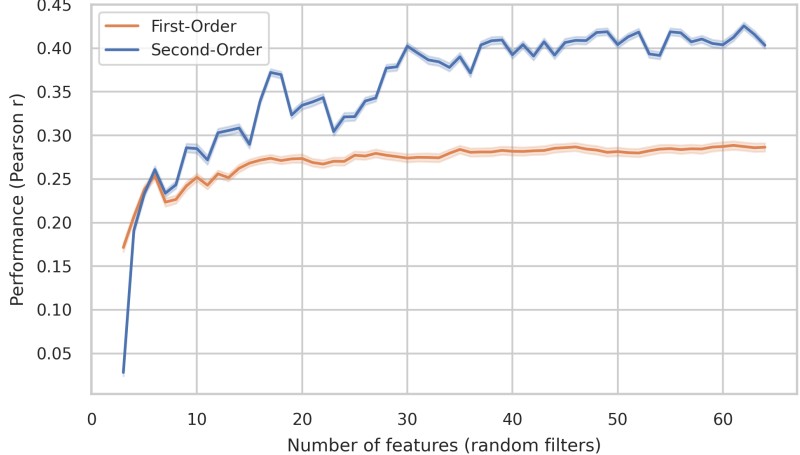

**Figure 4: Comparing first- and second-order encoding models with matched dimensionality.** We controlled for the increase in dimensionality when taking pair-wise feature interactions by using a random subset of interactions to fit the encoding model, with the number of interactions matched to the number of first-order features. Using the random-weight feature extractor and varying the number of filters used, second-order models consistently outperformed first-order ones in predicting fMRI data from visual cortex, even when their dimensionality was matched. Lines represent the mean voxelwise correlation between predicted and actual fMRI responses across EVC, object, scene, and face regions in a cross-validation design. Shaded regions represent the 95% confidence intervals computed from bootstrap distributions.

## 3  Discussion

Here we investigated the hypothesis that multiplicative feature interactions serve as an important canonical computation in human visual cortex. Using a simple, parameter-less operation that computed second-order multiplicative interactions between CNN feature maps, we found that we were able to significantly improve the prediction accuracy of fMRI encoding models across visual cortex. This improvement was consistently observed across all brain regions and CNNs considered, providing evidence in favour of our hypothesis.

Several questions remain regarding the nature of multiplicative interactions and the conditions under which they are effective in explaining neural data. Here, we limited our investigation to one specific type of multiplicative interaction: pair-wise products aggregated across space. However, higher-order interactions and operations that preserve spatial information might further improve encoding performance. Furthermore, additional work is needed to determine the underlying properties of multiplicative interactions that make them useful and efficient predictors of the representations in visual cortex. It is possible that similar results could be obtained through other means of nonlinear dimensionality expansion (Babadi & Sompolinsky, 2014; Cayco-Gajic & Silver, 2019). This would suggest that multiplicative interactions belong to a broader class of dimensionality-expansion algorithms that are highly effective at predicting representations in high-level visual cortex. We are currently investigating this possibility.

Recently, the field of computational neuroscience has had tremendous success in using deep learning to model human cortical representations and cognition (Bashivan, Kar, & DiCarlo, 2019; Khaligh-Razavi & Kriegeskorte, 2014; Ponce et al., 2019; Richards et al., 2019; Storrs, Kietzmann, Walther, Mehrer, & Kriegeskorte, 2020; Yamins & DiCarlo, 2016; Yamins et al., 2014). This success, however, has come at the cost of a diminished interest in other approaches with a potential to make parallel breakthroughs, such as an investigation into nonlinear canonical neural computations. Indeed, we saw here that one such class of computations, multiplicative feature interaction, dramatically improved various existing models of visual cortex. We also observed, however, that multiplicative interactions were most effective when applied to features extracted by deep CNNs trained with backpropagation. This serves as a reminder that different, parallel approaches in computational neuroscience can have synergistic explanatory power, and that important insights can be missed if the field puts too much emphasis *exclusively* on deep learning.

In this paper, we presented empirical results on multiplicative interactions, but did not deeply investigate their computational properties or explore theories pertaining to their purpose in the brain. These will be topics of future work in our lab, and luckily there are interesting leads in the existing literature. Jayakumar et al. (2020) discussed multiplicative interactions in the context of deep learning as an important inductive bias that could improve model expressivity. Our second-order interaction layer also amounts to a computation of the Gram matrix, which has been used as a powerful representation of mid-level joint feature statistics in neural style transfer and texture synthesis (Gatys, Ecker, & Bethge, 2015; Li, Wang, Liu, & Hou, 2017), and it may be possible to bring the theoretical insights from those works under the larger umbrella of multiplicative interactions. Similarly, it would be informative to explore the functional similarities and differences between multiplicative interaction and divisive normalization, which has also been proposed as a canonical neural computation (Carandini & Heeger, 2012). Finally, multiplicative interactions may have important theoretical implications for investigators seeking human-interpretable models of feature tuning in mid- to high-level vision. Namely, if visual representations rely on highly complex multiplicative interactions, finding compact and intuitive descriptions of feature representations may be extremely challenging, and perhaps fruitless in many cases (Lillicrap & Körding, 2019).

## Broader Impact

We do not believe that considerations of broader impact are applicable to this work.

## Acknowledgments and Disclosure of Funding

We thank Donald Li for sharing code for the curvature convolutional filters. We also thank our reviewers for providing valuable feedback and suggesting future research directions to explore.

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

# Appendix A  Feature extraction

All of our feature extraction models were CNNs composed of a sequence of linear convolutions, spatial max-pooling operations, and rectified linear (ReLU) nonlinearities. We experimented with both hand-engineered models and supervised ones trained on classification tasks.

**Hand-engineered models.**   We created three sets of hand-engineered convolutional filters. The Edge model contains oriented edge detectors created from Gabor filters, which were inspired by known tuning properties of V1 (Olshausen & Field, 1997). The Curvature model contains curved-contour detectors that were created by combining a rotated and curved complex wave function and a rotated and curved Gaussian function. The Curvature model was inspired by findings indicating that curvature is an important property of mid-level representations in primate visual cortex (Long, Yu, & Konkle, 2018; Yue, Pourladian, Tootell, & Ungerleider, 2014; Yue, Robert, & Ungerleider, 2020), and our convolutional filters are similar to those used in previous neuroscience studies (Yue et al., 2014). The Random model was created by generating random convolutional filter weights between 0 and 1 and mean-centering each filter. Precise architectural details and filter banks for each of these models are provided in Appendix G.

**Supervised models.**   In addition to these hand-engineered models, we also considered 2 deep CNNs that were trained on a supervised classification task. Our first CNN was trained on the CIFAR100 dataset and consisted of 3 convolution/ReLU/max pool layers that took 32x32 $RGB$ images as input and output 64 1x1 feature maps. Our second CNN was trained on a subset of the ImageNet dataset and had an architecture that mirrored the feature extraction layers of AlexNet (Krizhevsky, Sutskever, & Hinton, 2012), except all channel sizes were reduced by a factor of 4, resulting in an output of 64 6x6 feature maps from 224x224 $RGB$ images. The primary reason for reducing the number of channels across the model was to preserve most of the architecture, which is commonly used in the cognitive neuroscience literature, while reducing its predictive power as a computational model of visual cortex. In particular, an encoding model trained using features from an ordinary AlexNet was already approaching the noise ceiling of our fMRI dataset, making it difficult for us to assess any potential improvement in performance resulting from using second-order interactions of those same features. Further details about training methodology and datasets are included in Appendix D, and precise layer-by-layer descriptions of the above architectures are provided in Appendix G.

# Appendix B  Second-order feature interactions

The specific multiplicative feature interactions that we considered in this work were pair-wise products between each of the feature maps output from a CNN. Consider two such feature maps, $F^i$ and $F^j$, both matrices of shape $(H, W)$. To get the multiplicative interactions between these matrices at each spacial location, we first computed their element-wise product, also known as the Hadamard product. We then aggregated these multiplicative interactions by taking their sum across all spatial dimensions. The multiplicative interaction between $F^i$ and $F^j$ is therefore given by the equation:

$$M^{ij} = \sum_{h=1}^{H} \sum_{w=1}^{W} F_{hw}^i F_{hw}^j$$

If we flatten the feature maps such that they become vectors of length $HxW$, then the above equation simply denotes the inner product between $F^i$ and $F^j$. Therefore, if we are interested in the pair-wise multiplicative interactions between $n$ feature maps, we can compute them all at once by first flattening and concatenating each feature map to produce a matrix of shape $(n, HxW)$, then multiplying it by its transpose to produce a multiplicative interaction matrix $M$ of shape $(n, n)$. Finally, because $M$ is symmetric, we can keep only the upper triangular section and ignore the diagonal, so that the final result is a vector $\mathbf{x}^{O(2)}$ of length $\binom{n}{2}$ containing the second-order multiplicative interactions between all pairs of feature maps summed across spatial dimensions.

Since the above operation aggregates second-order feature interactions across spatial dimensions, we performed a similar aggregation of first-order features by taking the spatial mean of the feature maps, resulting in a vector $\mathbf{x}^{O(1)}$ of length $n$.

Prior to computing the second-order interactions, we also normalized feature map activity across the channel dimension. In particular, at each spatial location, we computed the mean and standard deviation across features and then normalized them such that the new mean was 0 and the new standard deviation was 1. We performed this normalization because we empirically observed that it improved the performance of our encoding models, but on its own it did not account for the results. In other words, across-channel normalization enhanced the effectiveness of multiplicative interactions in predicting fMRI data, but was not the driving factor.

We also note that our lab is currently working on improved operations for computing multiplicative feature interactions that a) preserve spatial information and b) learn which feature interactions are important so that dimensionality does not increase quadratically.

## Appendix C    Encoding model

Our final goal was to compare first-order features $\mathbf{x}^{O(1)}$ to their second-order interactions $\mathbf{x}^{O(2)}$ with respect to how well they predict fMRI activity across visual cortex. To this end, we trained encoding models on an fMRI dataset described in Bonner and Epstein (2020) with data across 4 subjects. The stimulus set consisted of 810 objects from 81 different categories (10 object tokens per category). Example stimuli are shown in Appendix H. fMRI responses were measured while subjects viewed these objects, shown alone on meaningless textured backgrounds, and performed a simple perceptual task of responding by button press whenever they saw a "warped" object. Warped objects were created through diffeomorphic warping of object stimuli.

Using a voxelwise modeling procedure, we examined to what extent fMRI responses to these object stimuli could be predicted from the features extracted by the CNNs described in Appendix A or their second-order interactions described in Appendix B. We first estimated the fMRI responses to each object category. We then fit voxelwise encoding models with the goal of predicting the fMRI responses to all object categories through a weighted linear sum of the first- or second-order features. Specifically, a set of linear regression models were estimated using the first- or second-order features as regressors and the voxelwise fMRI responses as predictands, as shown in Figure 5. Through cross-validation, we assessed how well the estimated encoding models could predict fMRI responses to out-of-sample stimuli. The cross-validation procedure was designed so that the training and test sets always contained objects from different categories, which allowed for a strong test of generalization to new semantic categories, rather than new stimuli from the same categories.

Finally, we performed a series of region of interest (ROI) analyses so that we could assess the predictive value of second-order feature interactions in multiple brain areas associated with different visual functions. These included early visual cortex (EVC) as well as higher-level regions associated with object, (LOC and PFS), scene, (PPA and OPA), and face (FFA and OFA) processing. Each ROI consisted of roughly 100 voxels per subject.

## Appendix D    Supervised model training

We performed our experiments on two models trained to perform image classification tasks. Here, we first describe aspects of the training pipeline that were common to both models, and then discuss each model's architecture and dataset individually.

**Training pipeline.**    We trained our models using backpropagation with the Adam optimizer (Kingma & Ba, 2014) to minimize the cross-entropy loss over predicted image classes. We used a constant learning rate of $1 \times 10^{-4}$ and a batch size of 64. All training was done using PyTorch (Paszke et al., 2019). Following the feature extraction backbones of our networks, we computed either the first-order feature vector $\mathbf{x}^{O(1)}$ or the vector of second-order feature interactions $\mathbf{x}^{O(2)}$. These vectors were then followed by a number of fully-connected layers with ReLU nonlinearities before finally being mapped to a vector of class predictions. For all models, we saved only the set of parameters that resulted in the lowest cross-entropy loss on the test set.

**CIFAR100 CNN.**    Our first supervised model was trained on the CIFAR100 dataset, which consists of 100 classes each containing 600 32x32 $RGB$ images (Krizhevsky, 2009). The dataset was split into a training set of 500 images per class and a test set of 100 images per class. We trained the model for a total of $5 \times 10^4$ iterations.

**ImageNet CNN.**    Our second supervised model was trained on a truncated version of the ImageNet dataset (Deng et al., 2009), which we constructed by randomly sampling 1000 categories with 1000 224x224 $RGB$ images per category. The dataset was split into a training set of 900 images per class and a test set of 100 images per class. We trained the model for a total of $5 \times 10^5$ iterations.

## Appendix E    Second-order encoder and noise ceiling

We observed that our best second-order encoding model, derived from a supervised CNN pre-trained on ImageNet, approached the noise ceiling of our fMRI dataset, as shown in Figure 6.

## Appendix F    Comparisons to a conventional AlexNet

Currently, deep neural networks trained on object classification tasks are the leading models of high-level visual areas (Storrs et al., 2020). We therefore compared the performance of our second-order models to

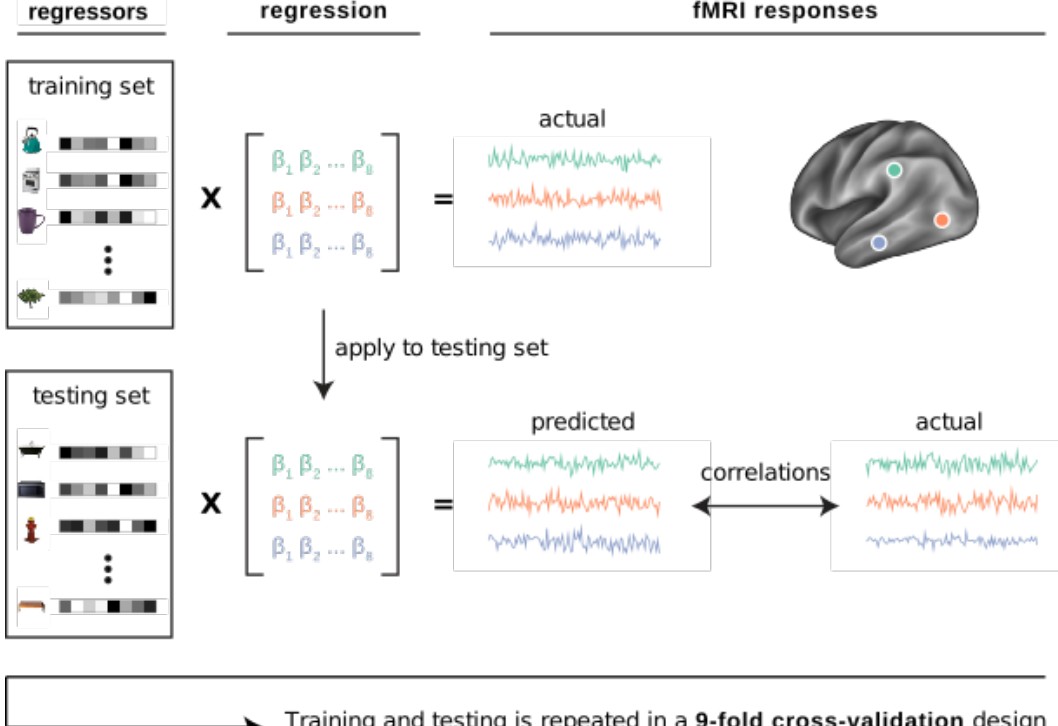

Training and testing is repeated in a **9-fold cross-validation** design

**Figure 5: Encoding model training.** Voxelwise encoding models were used to assess to what extent our first- and second-order feature sets could reliably explain variance in the fMRI responses to the experimental object categories from the original dataset. Linear regression was used to map the first- and second-order representations onto fMRI responses. We assessed the out-of-sample prediction accuracy of these regression models through a 9-fold cross-validation procedure. Each fold of the cross-validation design contained a set of object categories that did not appear in any other fold. These folds were shown in separate fMRI runs. Parameters for the voxelwise linear regression models were estimated using the fMRI data for 8 folds of the object categories and the learned regression weights were then applied to the held-out object categories in the remaining fold to generate a set of predicted fMRI responses. This procedure was repeated for each fold of the cross-validation design, and the predicted fMRI responses from each fold were concatenated together. Prediction accuracy was assessed by calculating the voxelwise correlations of the predicted and actual fMRI responses across all object categories.

that of a pre-trained AlexNet, which is a state-of-the-art deep CNN and a popular model of visual cortex in the computational neuroscience literature. In Figure 7, we see that several of our second-order models were competitive with AlexNet, while our first-order models comparatively under-performed. Moreover, our second-order ImageNet-trained CNN achieved the same predictive accuracy as AlexNet, but with an order of magnitude fewer parameters and significantly lower classification accuracy on ImageNet.

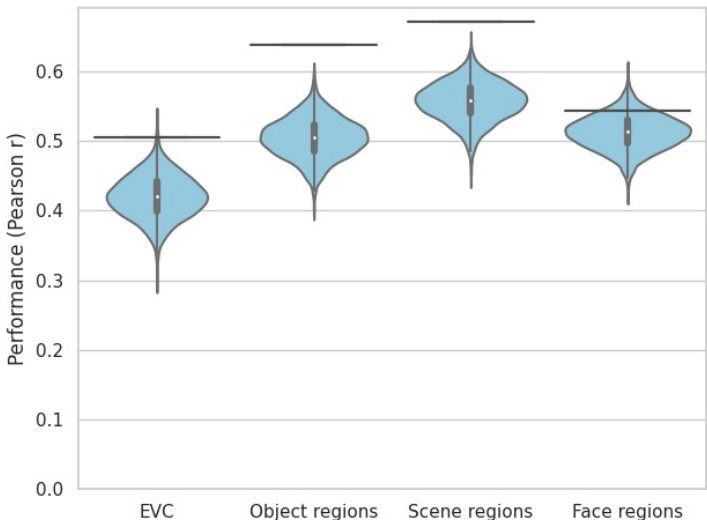

**Figure 6: Voxelwise encoding model of ImageNet-trained CNN second-order feature interactions compared to noise ceiling.** Second-order multiplicative interactions between ImageNet-trained CNN features were highly predictive of fMRI responses to objects in visual cortex, approaching the noise ceiling of the dataset (horizontal bars). The noise ceiling was computed as the average voxelwise split-half reliability of the fMRI responses in each visual region. Violin plots represent means and bootstrap distributions for each visual region.

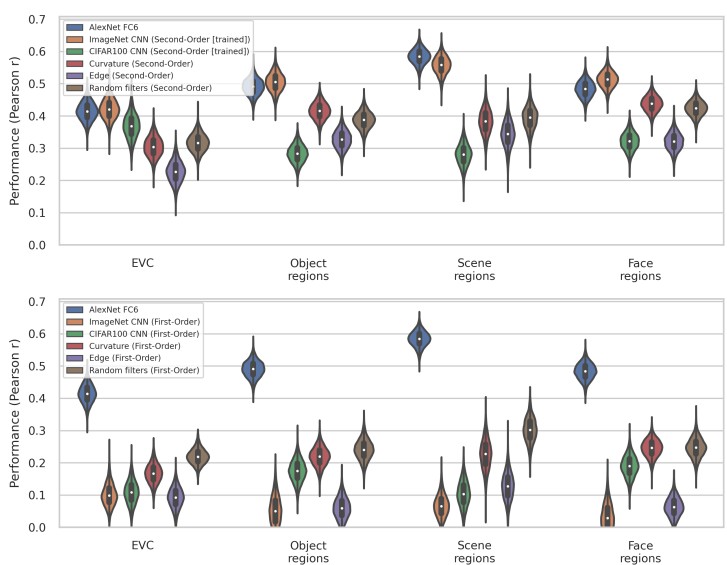

**Figure 7: Second-order models achieved similar encoding accuracy to a conventional pre-trained AlexNet. Top)** Second-order feature interaction models were competitive with AlexNet, a state-of-the-art deep CNN, in predicting fMRI responses across visual cortex. Our ImageNet-trained CNN, which was a reduced version of AlexNet with an order of magnitude fewer parameters, achieved the same encoding accuracy when second-order multiplicative feature interactions were used. **Bottom)** All of our first-order models performed significantly worse than AlexNet across all brain regions considered. Violin plots represent means and bootstrap distributions for each visual region. The output of AlexNet's first fully-connected layer was used to train the encoding model.

# Appendix G  Feature extraction layer-by-layer architectures

**Table 1:** Hand-engineered CNN architectures

| Input shape | Layers | Output shape |
|---|---|---|
| (H=96,W=96,C=1) | Conv(N=320,K=9,S=1,P=4), Abs | (H=96,W=96,C=320) |

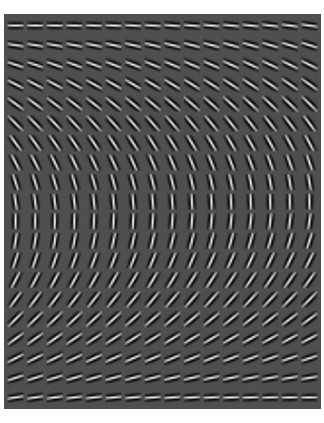 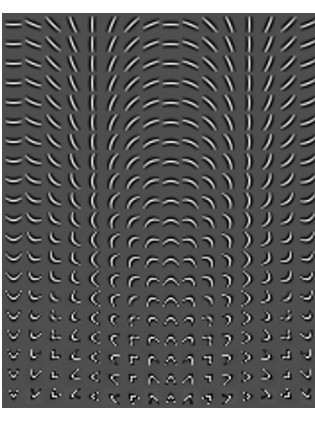 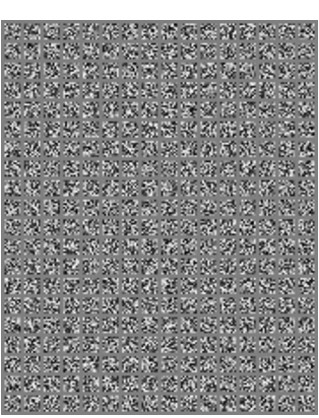

**(a)** Edge filter bank.  **(b)** Curvature filter bank.  **(c)** Random filter bank.

**Figure 8:** Filter banks for each hand-engineered CNN architecture.

**Table 2:** CIFAR100 CNN architecture

| Input shape | Layers | Output shape |
|---|---|---|
| (H=32,W=32,C=3) | Conv(N=16,K=7,S=1,P=0), ReLU, MaxPool(K=2,S=2) | (H=13,W=13,C=16) |
| (H=13,W=13,C=16) | Conv(N=32,K=5,S=1,P=0), ReLU, MaxPool(K=2,S=2) | (H=4,W=4,C=32) |
| (H=4,W=4,C=32) | Conv(N=64,K=3,S=1,P=0), ReLU, MaxPool(K=2,S=2) | (H=1,W=1,C=64) |

**Table 3:** ImageNet CNN architecture

| Input shape | Layers | Output shape |
|---|---|---|
| (H=224,W=224,C=3) | Conv(N=16,K=11,S=4,P=2), ReLU, MaxPool(K=3,S=2) | (H=27,W=27,C=16) |
| (H=27,W=27,C=16) | Conv(N=48,K=5,S=1,P=2), ReLU, MaxPool(K=3,S=2) | (H=13,W=13,C=48) |
| (H=13,W=13,C=48) | Conv(N=96,K=3,S=1,P=1), ReLU | (H=13,W=13,C=96) |
| (H=13,W=13,C=96) | Conv(N=64,K=3,S=1,P=1), ReLU | (H=13,W=13,C=64) |
| (H=13,W=13,C=64) | Conv(N=64,K=3,S=1,P=1), ReLU, MaxPool(K=3,S=2) | (H=6,W=6,C=64) |

# Appendix H    Sample stimuli for fMRI study

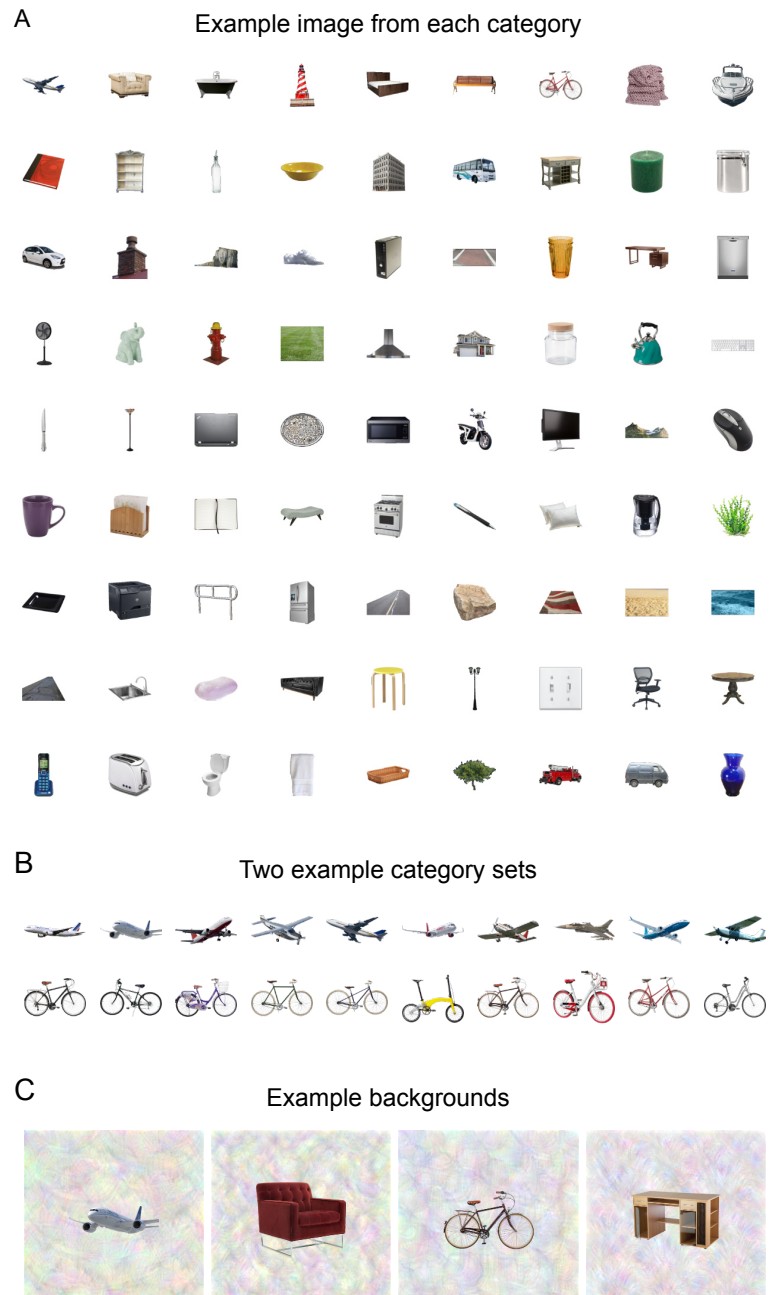

**Figure 9: Experimental stimuli used to train encoding models. A**) In an fMRI experiment, subjects viewed isolated images of objects from 81 different categories. This panel shows one example image from each category. **B**) There were 10 unique tokens for each category (for a total of 810 unique stimuli). This panel shows all 10 stimuli for two categories. **C**) In the fMRI scanner, images were presented on complex, textured backgrounds to reduce the saliency of low-level features related to object shape and size. This panel shows four example stimuli on textured backgrounds.

