# OpenReview forum: "Visual representations derived from multiplicative interactions"
_NeurIPS.cc/2020/Workshop/SVRHM — SVRHM@NeurIPS Poster_

### Official Review · AnonReviewer2 · 2020-10-29
**Interesting and comprehensive work but interpretation and results are oversold**

**Rating:** 6
**Confidence:** 4

**Review:**

In this work, the authors tested the hypothesis that multiplicative interactions of features are canonical computational units that provide better models of brain activity than using simple first order features. They used a variety of hand-crafted and learned models to extract features and compared the performance of using first-order vs. second order features extracted from these networks in order to predict voxelwise human fMRI responses to object presentations. They found that multiplicative interactions greatly increased the predictive power of the fMRI encoding models.


Pros:
* Overall, this work provides a set of nice experiments to test the hypothesis that including multiplicative interactions explicitly in neural encoding models can help in their predictive power. The results pretty clearly demonstrate that even controlling for dimensionality, second order features can improve on first order features in predicting voxelwise fMRI responses across multiple brain areas.
* The fact that this result holds true across a wide-variety of model architectures (hand-engineered and deep nets) (Fig. 6) is also encouraging and shows that this is not specific to any model type.
* Fig. 7 is also a very nice test to show that the multiplicative interaction is useful even when it is not trained, showing that the multiplicative interaction is something more fundamental than simply increasing model capacity during training.
* The writing is clear, ideas are thought through and the authors attempt to account for alternative explanations for their results.


Cons:
* The authors mention briefly that they had to reduce the complexity of AlexNet because AlexNet fc6 originally achieves noise-ceiling in the fMRI prediction. While I understand why the authors did this given their data, I think this is a bigger deal than they state in terms of larger impact of the work. Specifically, if the original pre-trained AlexNet reaches the noise ceiling in this neural prediction task, then how can it be claimed that multiplicative interactions are a necessary canonical computation. In comparison to the handicapped AlexNet (with lower dimensionality) this is true, but this is because the lower dimensionality AlexNet does not have the capacity to learn useful features on a complex dataset like ImageNet. The authors control for dimensionality in the comparison of first order and second order features (Fig 3), but this is in a regime where the models have restricted capacity relative to their training dataset. Once the capacity is increased to the original AlexNet size, is it actually still true that multiplicative interactions would help? Obviously the authors would need to find an fMRI dataset where the original AlexNet is not performing at noise-ceiling and then test this there (so I dont necessarily hold it against them), but I do think it tempers the overall claims. This is especially true when seeing that the handicapped model achieves 14% and 27% on ImageNet test accuracy (which is extremely poor in comparison to the original or really any other modern deep network)-- multiplicative interactions can clearly help a restricted model (relative to the training dataset) improve it's capacity , but if the dataset and model capacity are matched so that the model can already learn effectively (i.e. high accuracy on the task) then I'm not sure these results would be as clear.
* The authors don't seem to cite some of the literature that already uses multiplicative bilinear forms in CNNs to achieve second-order statistics and I think it would be worth it for them to take a look at these networks and compare the performance of these pre-trained state of the art networks that have second-order features to other comparable state of the art networks without second order features. To cite a few: "Bilinear CNN..." Lin et al. 2015; "Deep CNNs meet global covariance pooling.. " Wang et al. 2019;  Again, this would require finding a dataset where the fMRI prediction is not at noise-ceiling for a good pre-trained CNN but if this is doable, it would greatly improve the authors' case for these interactions being necessary even when the models have huge capacity.
* The authors mention "multiplicative interactions" generally, but in this work they focus on a very specific form (Gram matrix). It is still a bit unclear to me as to whether this specific form is what is providing the benefits, or whether similar results can be obtained simply by using other second order features. For example, do the authors get similar performance if they use the first-order features and squared features, removing the cross terms? I think that there is clearly a benefit to using the Gram matrix second-order features they propose, but I think the claims could be far more impactful if there was more comparisons to other higher-order feature pooling.
----------------------------------------------


In summary, I think the question and experiments are well thought out and it is clearly evident that as a parameter-free option, multiplicative interactions can immediately increase the neural predictive power of an otherwise impoverished model and this observation is generalizable across multiple brain areas.  Because of this, the hypothesis and preliminary results seem like something cool to build off of for future work.  That being said, I think in it's current state there is a non-trivial issue that the authors handicap the deep networks they use. Their claims that multiplicative interactions are actually a canonical computation may not be true when models have enough capacity to learn useful features as it is on their training data.  Because of this possibility, I think the current claims are overstated as a top-performing deep network with high capacity may achieve these benefits simply by having large enough dimensionality and enough layers of nonlinear transformations. Nevertheless I recommend acceptance for the ideas and analysis that has been done which shows promise.

---

### Official Review · AnonReviewer1 · 2020-10-29
**Interesting question and results -- and many paths for future inquiry.**

**Rating:** 7
**Confidence:** 4

**Review:**


### Motivation
In recent years, features learned in deep CNNs trained on image classification tasks have far surpassed classical neuroscience models at the task of predicting fMRI responses in human visual cortex. This has lead some researchers to consider the deep CNN architecture as a model _of_ visual cortex. There is a long history of connection between deep CNNs and models of visual cortex; the earliest convolutional and hierarchical neural networks drew heavy inspiration from the neuroscience literature. However, neuroscientists have long been aware that the basic linear-nonlinear unit used in CNNs is a vast simplification of biological neurons.

The authors of this paper start from this point, noting that the dynamics of many biological neurons involve multiplicative interactions between inputs. The authors ask: _Will features generated as pairwise multiplicative interactions of CNN features provide a better prediction of fMRI responses in human visual cortex than ordinary CNN features alone?_ The authors hope that this work may speak to the bigger question: _Is multiplicative interaction a canonical computation in human visual cortex?_

### Methods
The authors take the following approach:

1. Define a classical CNN architecture that maps images to features. This is the "first-order" model. The authors consider both learned weights (trained on Imagenet or CIFAR100), hand-engineered weights (e.g. Gabors and curves), and random weights.
2. Define a second-order model that computes all pairwise multiplicative interactions between the feature maps from the first-order model.
3. Collect fMRI data from humans viewing images of objects.
4. Train a linear model to predict the responses in different regions in visual cortex using either the first- or second- order model.
5. Compare the predictive accuracies of the two types of models.

### Results
The authors find that the features generated as second-order multiplicative interactions are better predictors of fMRI responses than the conventional features. The results hold in both lower and higher-level regions in visual cortex, and hold across all types of CNNs used, including those with hand-engineered weights and those trained on classification tasks.

### Challenges
The authors note two potential confounds:
1. The multiplicative models have more parameters than the conventional models. This could be an alternative explanation for their predictive power.
2. The multiplicative models were trained with the multiplicative layer as part of the model. This could affect the way the parameters were learned across lower layers in the network, and removes the guaranteed correspondence between the features learned in the first- and second-order models.

To address confound 1, the authors match the number of first-order features by using a random subset of the second-order interaction. These experiments are conducted with the random weight model and the authors varied the number of features used in both models. The authors find that the second-order model consistently outperforms the first-order one when matched in dimensionality. I believe this control satisfactorily dismisses this confound, but think this approach should be taken from the beginning.

To address confound 2, the authors conducted a new set of experiments in which second-order interactions are computed only after classification training. The authors find that the "untrained" second order model still performs better than the first-order model in most tasks, although it performs consistently worse than the trained model. I think the fact that the trained second-order model does considerably better is interesting in itself. If multiplicative interactions are indeed "canonical computations" for vision, then the visual cortex is more like the trained second-order model. But I will add that it was not clear in the body of the text that the model was trained with the multiplicative layer; it sounded like multiplicative interactions were computed on top of a pre-trained model. To make the trained CNN results comparable to the hand-engineered and random weight models, the authors should put these results in the main body rather than the appendix. Then, the model trained with the multiplicative layer can be considered as a second experiment. This confound clearly did effect the results, so it would be best to clear it up from the beginning.

I'd like to note a couple additional confounds:

3. In the first-order model, the feature maps are flattened and then the pixel mean is taken per feature. The mean operation destroys much of the information in the feature map, and it is unclear why this is necessary. I suppose it could be an effort to match the second-order method, but I'd like to understand the authors' reasoning behind this choice. The authors may get better results from the first-order model if the predictive model were trained on the raw features themselves.

4. As the authors note, any nonlinear continuous function, including pairwise multiplicative interaction, can be approximated with a piecewise linear function. A deep network can in principle learn to output multiplicative features, if there are enough neurons and layers. A multiplicative interaction layer is a way to get more complexity from fewer neurons. These multiplicative features may be a better fit to the fMRI data because they are more complex, and the ReLU models simply fail to learn those features with the number of neurons and layers given. One could argue that the results do not show that multiplicative interactions are specifically important, but rather that increasing model complexity gives a better fit. It would be good to see the authors test multiple hypotheses, to narrow in on the primary driver of these results. Are pairwise interactions specifically important? How about three-way interactions, or other types of nonlinearities? If you keep increasing the size of the ReLU networks, can you match the performance of the multiplicative nets? A thorough investigation of the space of possible canonical computations would be very valuable and interesting.

Some other things I would like to see:
- The mathematics of the second-order layer should be explicitly written.
- The paper would benefit for breaking up the Results section into Methods and Results.

---

### Official Review · AnonReviewer3 · 2020-10-30
**Interesting experiment**

**Rating:** 7
**Confidence:** 3

**Review:**

The authors present evidence supporting their hypothesis that multiplicative interactions play a significant role in visual processing. They use handcrafted and ANN-based representations trained on image classification to predict fMRI data, showing that multiplicative computations added to the representation increase predictive power.

I’ve found this quite an interesting paper, and was initially skeptical due to the concern about matching dimensionalities, which the authors address satisfactorily. Some doubt remains, as this is fMRI data, and as such not directly a representation, but merely a localized indicator of neural activity.

I wonder if an alternative explanation for the results could be that both the multiplicative responses as well as the fMRI data can be interpreted as an energy of some sort (the former as quadratic expressions akin to second moments). Hence, a linear predictor would be more successful in predicting one from the other. In other words, fMRI is a measure of neuronal activity, the gram matrix is a measure of activity of artificial neurons - perhaps this explains why they correlate better.

---

### Public Comment · ~Eric_Elmoznino1 · 2020-11-26
**Author reply**

We would like to thank all reviewers for their helpful and constructive comments. Based on their feedback, we have made the following changes to our paper:
- We previously reported in the Appendix the results of supervised CNNs that were pre-trained without the second-order layer. We have now moved these results to the main text and report them along with the results obtained when including the second-order layer during pre-training.
- We now include the formula for the second-order interaction layer in the main text, as opposed to only in the Appendix.
- In the Discussion, we clarify that the specific second-order layer we explored is not the only way one could compute multiplicative interactions, and that other schemes (e.g. triplet interactions) could potentially provide better performance. We note that we are currently conducting a more comprehensive study into the various forms of multiplicative interactions.
- We are currently performing analyses to compare multiplicative interactions with other approaches for nonlinear dimensionality expansion. We have commented on the implications of this in the Discussion.

---

### Decision · Program_Chairs · 2020-11-02

Accept (Poster)